# Prevalence of Carbapenemases in Carbapenem-Resistant *Acinetobacter baumannii* Isolates from the Kingdom of Bahrain

**DOI:** 10.3390/antibiotics12071198

**Published:** 2023-07-17

**Authors:** Nouf Al-Rashed, Khalid M. Bindayna, Mohammad Shahid, Nermin Kamal Saeed, Abdullah Darwish, Ronni Mol Joji, Ali Al-Mahmeed

**Affiliations:** 1Department of Microbiology, Immunology, and Infectious Diseases, College of Medicine & Medical Sciences, Arabian Gulf University, Manama P.O. Box 26671, Bahrain; bindayna@agu.edu.bh (K.M.B.); mohammeds@agu.edu.bh (M.S.); ronnimj@agu.edu.bh (R.M.J.); aliem@agu.edu.bh (A.A.-M.); 2Department of Pathology, Microbiology Section, Al- Salmaniya Medical Complex, Manama P.O. Box 12, Bahrain; nhasan@health.gov.bh; 3Department of Pathology, Microbiology Section, Bahrain Defense Force Hospital, West Riffa P.O. Box 28743, Bahrain; abdulla.darwish@bdfmedical.org

**Keywords:** *Acinetobacter baumannii*, carbapenemases, OXA, KPC, NDM, IMP, VIM

## Abstract

Background: *Acinetobacter baumannii* is regarded as a significant cause of death in hospitals. The WHO recently added carbapenem-resistant *Acinetobacter baumannii* (CRAB) to its global pathogen priority list. There is a dearth of information on CRAB from our region. Methods: Fifty CRAB isolates were collected from four main hospitals in Bahrain for this study. Bacterial identification and antibiotic susceptibility tests were carried out using the BD Phoenix^TM^ and VITEK-2 compact, respectively. Using conventional PCR, these isolates were further screened for carbapenem resistance markers (*bla_OXA-51_, bla_OXA-23_, blaO_XA-24_, bla_OXA-40_, bla_IMP_, bla_NDM_, bla_VIM_,* and *bla_KPC_*). Results: All of the isolates were resistant to imipenem (100%), meropenem (98%), and cephalosporins (96–98%), followed by other commonly used antibiotics. All these isolates were least resistant to gentamicin (64%). The detection of resistance determinants showed that the majority harbored *bla_OXA-51_* (100%) and *bla_IMP_* (94%), followed by *bla_OXA-23_* (82%), *bla_OXA-24_* (46%), *bla_OXA-40_* (14%), *bla_NDM_* (6%), *bla_VIM_* (2%), and *bla_KPC_* (2%). Conclusion: The study isolates showed a high level of antibiotic resistance. Class D carbapenemases were more prevalent in our CRAB isolate collection. The resistance genes were found in various combinations. This study emphasizes the importance of strengthening surveillance and stringent infection control measures in clinical settings to prevent the emergence and further spread of such isolates.

## 1. Introduction

*Acinetobacter baumannii* (*A.baumannii*) is emerging as a significant multidrug-resistant (MDR) pathogen in hospitals, particularly in intensive care units (ICUs), and it is considered a major nosocomial pathogen causing high mortality [1,2]. The reported mortality rate is around 7.8% to 23% in hospitals and around 10% to 43% in ICUs [3]. Although there has not been any clear consensus on the associations between carbapenem-resistant *Acinetobacter baumannii* (CRAB) infections and an elevated risk of mortality [4], CRAB infections have shown a significant correlation with the length of ICU stays, elevated patient costs, and antibiotic use [4]. Moreover, it is also considered a significant pathogen causing hospital-acquired infections (HAIs) that increase the risk of the emergence of pan-drug resistance and outbreaks [5]. It usually infects human skin and wounds, especially the respiratory, gastrointestinal, and circulatory systems, causing serious infection [6]. Examples of HAIs are bacteremia, septicemia, wounds, meningitis, ventilator-associated pneumonia, and urinary tract infections [6]. Countries in the Mediterranean area have some of the highest resistance rates to carbapenems on *A. baumannii*, reaching 90%, including the Middle East, southern Europe, and North Africa [7]. The countries with the most MDR *A. baumannii* infections in the Middle East are the United Arab Emirates, Bahrain, Saudi Arabia, Palestine, and Lebanon [6]. Another epidemiological study in 2019 revealed that the resistance rates in Asia-Pacific, East Asia, Europe, North America, and Latin America were 56%, 100%, 60%, 36%, and 54%, respectively [8]. Furthermore, community-acquired pneumonia infections can also occur in several tropical countries (e.g., Asia and Australia) because of high levels of rain and humidity [6]. Owing to its significance, the World Health Organization has included CRAB in its global priority list of pathogens.

*A. baumannii* infections are thought to affect 1 million people worldwide each year, with 50% of those cases developing resistance to various medicines, including carbapenems [9]. Resistance is the outcome of multiple systems acting simultaneously and in unison. Specifically, these include (a) the lack and small size of outer-membrane porins whose expression can be further reduced, (b) the constitutional expression of efflux pumps (AbeABC, AbeFGH, and AbeIJK), (c) certain β-lactamases’ intrinsic expression (carbapenemases, AmpC cephalosporinases, and OXA-like β-lactamases), (d) the occurrence of a ‘resistance island’, and (e) the ability for the horizontal acquisition of resistance determinants (OXA-23 and NDM carbapenemases, as well as aminoglycoside-modifying enzymes) [10]. Most CRAB isolates are extensively drug-resistant (XDR), indicating that they are not susceptible to antibiotic classes other than polymyxins and tigecycline [10]. Nevertheless, colistin and/or tigecycline-resistant CRAB strains are being reported more frequently [10]. As a result, a large number of clinical isolates are pan-resistant [10].

The resistance rate of *A. baumannii* to carbapenems has even reached 90% in the Middle East, southern Europe, and North Africa [7]. Most of the *A. baumannii* infections in Middle Eastern countries were reported in the United Arab Emirates, Saudi Arabia, Palestine, and Lebanon [6]. In these circumstances, it is possible that the control and treatment of CRAB will lead to new difficulties, which have sparked considerable concern in the medical community [11].

The goal of this study was to determine carbapenemase production in CRAB by identifying the specific type of *bla*-carbapenemase genes (and their prevalence) in our collection of CRAB isolates collected from four major hospitals in the Kingdom of Bahrain. Furthermore, the outcomes of gene prevalence and antibiotic susceptibility patterns were tested to discover the similarities or differences between the GCC region and the international region.

## 2. Results

### 2.1. Distribution and Antibiotic Resistance Pattern of the Isolates

The CRAB isolates used in this study were obtained from endotracheal aspirates (Number (*n*) = 16), blood (*n* = 11), urine (*n* = 7), wound swabs (*n* = 7), pus swabs (*n* = 4), rectal swabs (*n* = 3), and sputum (*n* = 2). These isolates showed significant resistance to imipenem (*n* = 50/50, 100%), meropenem and cefuroxime (*n* = 49/50, 98%), cefepime, cefotaxime and ceftriaxone (*n* = 48/50, 96%), ceftazidime (*n* = 45/50, 90%), amikacin (*n* = 21/40, 52.5%), gentamicin (*n* = 32/50, 64%), ampicillin/sulbactam (*n* = 18/29, 62.06%), ciprofloxacin (*n* = 46/50, 92%), levofloxacin (*n* = 42/50, 84%), ertapenem (*n* = 27/28, 96.4%), piperacillin/tazobactam (*n* = 41/44, 93.18%), trimethoprim/sulfamethoxazole (*n* = 28/47, 59.5%), minocycline (*n* = 11/30, 36.6%), tigecycline (*n* = 3/35, 8.5%), and colistin (*n* = 3/16, 18.75%). Details of the antibiotic resistance patterns of each isolate are shown in Figure 1A,B. The frequency of resistance of the respective antibiotics in the CRAB isolates is shown in Table 1.

### 2.2. Carbapenemase-Encoding Genes’ Detection

Among the tested class D carbapenemases, the *bla_OXA-51_* was detected in all 50 (100%) isolates. The *bla_OXA-23_* was detected in 41 (82%) isolates, followed by *bla_OXA-24_* in 23 (46%) isolates. Seven (14%) isolates showed the presence of *bla_OXA-40_***.** PCR was negative for *bla_OXA-48_* and *bla_OXA-58_* in all the isolates.

Among the tested Class B carbapenemases, *bla_IMP_* was detected in 94% of the isolates (*n* = 47), and *bla_NDM_* was detected in 6% (*n* = 3) of the isolates. *bla_VIM_* and *bla_KPC_* were detected in one isolate each. Detailed results are presented in Figure 1. Various combinations of genes were noticed in our collection of CRAB isolates, ranging from as few as two genes to as many as six genes in the respective isolates (Figure 1). The majority (44%) of the isolates had a combination of three genes (*bla_OXA-51_*, *bla_OXA-23_*_,_
*bla_IMP_*), followed by a combination of four genes (*bla_OXA-51_*, *bla_OXA-23_*_,_
*bla_OXA-24_*, and *bla_IMP_*) in 12 isolates (24%). The detailed results are presented in Table 2. A representative PCR gel demonstrating the respective amplicons is shown in Figure 2.

## 3. Discussion

*A. baumannii* is a pathogen of concern worldwide in the context of nosocomial infections owing to its multidrug resistance, often including drugs of last resort such as carbapenems [4]. This is a plausible reason why the WHO has included CRAB in its global priority list. Of the various mechanisms of carbapenem resistance in CRAB, the production of carbapenem-hydrolyzing enzymes is one of the main mechanisms of resistance [4]. These enzymes are mainly produced by the genes encoding carbapenem-hydrolyzing enzymes [4].

It is interesting to note that the Middle East was historically linked to *A. baumannii*, often known as “Iraqibacter,” due to an epidemic of resistant strains among the US military during the Iraq War. Since then, hospitals around the Middle East, including those in the United Arab Emirates, Saudi Arabia, Bahrain, Palestine, and Lebanon, have isolated this bacteria [6]. *A. baumannii* has been subjected to numerous investigations testing its susceptibility to various antibiotic classes. According to a study from the holy cities of Saudi Arabia, the screening of carbapenem-resistant *A. baumannii* isolates revealed that imipenem and meropenem resistance was widespread in 81% and 84% of the strains, respectively, while the majority of the organisms were colistin- and tigecycline-susceptible [12]. Another study from Saudi Arabia concluded that eight isolates (30%) were resistant to colistin, 15 isolates (56%) were resistant to tigecycline (56%), and 24 isolates (89%) were resistant to one or more carbapenems (imipenem and meropenem] [13]. The present study showed significant resistance to imipenem, meropenem, and cephalosporins and comparatively lower resistance to minocycline, tigecycline, and colistin. Similarly, a study in China reported a high resistance rate to carbapenems and cephalosporins and a lower resistance rate to levofloxacin, minocycline, and tigecycline [14]. A two-year retrospective study from Saudi Arabia also documented that almost all isolates of *A. baumannii* were carbapenem-resistant (98%). It was interesting to note that these isolates had higher resistance to colistin (15%) when compared to tigecycline (3%) [15]. These high levels of resistance rates were caused by the overuse of imipenem and meropenem for the treatment of *A. baumannii* infection in patients [16]. As a result, recommendations for the administration of infection control procedures are required to curb the spread of these isolates in hospital settings [16].

To date, we believe that no study has determined the prevalence of carbapenemase genes from Bahrain in a relatively large cohort of CRAB isolates. Earlier, in 2015, a collaborative effort was performed in a published joint research paper incorporating *A. baumannii* isolated from the Gulf Cooperation Council (GCC) countries. Most isolates were collected from Saudi Arabia, whereas a small proportion were collected from other GCC countries (only eight isolates were gathered from Bahrain hospitals) [17]. In that study, the researchers collected a total of 117 CRAB isolates from six countries (mainly Saudi Arabia) and reported the presence of *bla_OXA-51-type_* in all the isolates (100%; 117/117) and that of *bla_OXA-23-type_* in 91% (107/117) of the isolates [17]. Cumulatively, *bla_OXA-40-type_* was detected in 4% (5/117) of the isolates; all five isolates positive for this gene type were from Bahrain, and none of the isolates from other GCC countries demonstrated this gene type. Among the Bahraini isolates, all eight (100%) demonstrated the presence of *bla_OXA-51-type_*, followed by *bla_OXA-40-type_* (62.5%), and three isolates (38%; 3/8) showed *bla_OXA-23-type_* [17]. The authors also reported the non-detection of *bla_OXA-58_*, *bla_KPC_*, and metallo-beta-lactamases (MBLs) like *bla_IMP_* and *bla_VIM_* [17].

In another report published based on a study in Bahrain in 2009, where eight isolates were again molecularly tested for these carbapenemase genes [18], the most prevalent reported gene was *bla_OXA-40-like_* (in five isolates), followed by *bla_OXA-23_* (two isolates) and *bla_OXA-58_* (one isolate) [18].

In the current study, *bla_OXA-51_*, *bla_OXA-23_*, *bla_OXA-24_,* and *bla_IMP_* were the most commonly detected carbapenemase-producing genes, occurring at frequencies of 100%, 82%, 46%, and 94%, respectively. In contrast to the previous reports, our collection of CRAB isolates also showed the presence of *bla_VIM_*, *bla_NDM_*, *bla_KPC_*, and *bla_OXA-40_*, though with lesser frequency. None of our isolates showed the presence of *bla_OXA-48_* or *bla_OXA-58_*. Even though it is too early to speculate on the present context due to the small number of isolates tested previously, it looks as if the molecular epidemiology in Bahrain has changed over the years, with the predominant gene now being *bla_OXA-23_*, which was comparatively less prevalent earlier. The *bla_OXA-40_* gene has become less prevalent, being the predominant gene reported in previous studies. It is also alarming to note the presence of a combination of carbapenem-resistance genes in some isolates at a level as high as six genes. MBLs and *bla_KPC_* were rarely reported in *A. baumannii* isolates, except for *bla_IMP_*. However, in our isolates, we found the presence of *bla_IMP_* in a significantly higher percentage (94%), which is also alarming. It was noticed that there was no correlation between the combinations of carbapenemase genes and the antibiotic resistance pattern. The CRAB isolates were highly resistant to all carbapenems and most cephalosporins with lower resistance to other antibiotics.

The *bla_OXA-51_* is generally found in all *A. baumannii* isolates, being either carbapenem-resistant or carbapenem-sensitive isolates [19,20,21]. This gene is not associated with resistance unless insertion sequences (*ISAbaI*) upstream of *bla_OXA-51_* are involved, causing over-expression leading to carbapenem resistance, especially to imipenem [21,22]. Therefore, it is recommended as an excellent marker for species identification but not as a resistance marker [22,23,24,25].

On the other hand, *bla_OXA-23_* is a significant cause of carbapenem resistance in *A. baumannii* [19,21,22]. It has been reported as a prevalent gene in various studies published in several countries, including Saudi Arabia and Iran [19,21]. In contrast to this, a few other international studies have observed the presence of *bla_OXA-23_* at a lower frequency, as reported in Bosnia, Poland, and Croatia [21]. The *bla_O__XA__-24_* gene is also reported as a common gene, albeit at variable percentages [19,22,23]. On the Arabian Peninsula, another study from Egypt investigated the prevalence of carbapenemase genes in 40 CRAB isolates [26]. The *bla_OXA-51_* gene was amplified in all isolates, whereas *bla_OXA-23_*, *bla_OXA-24_*, and *bla_OXA-58_* were present in 50%, 7.5%, and 5% of the isolates. All these isolates lacked *bla_KPC_* or MBLs [26].

In a study conducted in Iran in 2015, Azizi O et al. observed that *bla_OXA-51_* and *bla_OXA-23_* were present in all isolates but were negative for *bla_OXA-58_* [19]. In South Africa, Lowings M. et al. reported two genes (*bla_OXA-51_* and *bla_OXA-23_*) among 100 MDR *A. baumannii* isolates (99 and 77%, respectively) [25]. The other genes, such as *bla_OXA-24_*, *bla_OXA-58_*, *bla_KPC_*, and MBLs, were negative [25]. Various other international studies outside of the GCC region also reported the presence of these genes, albeit with varying frequencies [24,27]. However, it is interesting to note that many of these studies reported the absence of genes such as *bla_OXA-24_*, *bla_OXA-58_*, *bla_IMP_*, *bla_VIM_*, and *bla_KPC_* [27].

Alarmingly, a significant proportion of the isolates in our collection also demonstrated concomitant resistance to fluoroquinolones and the aminoglycoside group of antibiotics. For other last-resort antibiotics such as tigecycline and colistin, even though the isolates demonstrated a lower frequency of resistance, the appearance of resistance is quite alarming.

This study has a few limitations. One is that the analysis of resistance determinants using molecular methods was limited to carbapenemases in CRAB isolates and did not include the details of ESBL and other antibiotic resistance mechanisms. In addition, sequencing was not performed to search for gene mutations. However, to the best of our knowledge, this is the first report describing the prevalence (and molecular characterization) of CRAB isolates in a relatively large cohort from the Kingdom of Bahrain.

## 4. Materials and Methods

### 4.1. Bacterial Isolates and Hospital Setting

From February 2021 to June 2022, 50 random, nonrepetitive CRAB isolates were collected from the microbiology labs of four different hospitals (Al-Salmaniya Medical Complex, Bahrain Defense Force Hospital, King Hamad University Hospital, and Bahrain Specialist Hospital) in the Kingdom of Bahrain. These isolates were cultured from specimens such as endotracheal aspirates, urine, sputum, blood cultures, wounds, pus, and rectal swabs. The isolates obtained from the lab were preserved in glycerol milk at certain volumes (3 mL and 4 mL) with 13 mL of deionized water and stored at −80 °C until further testing [28].

### 4.2. Bacterial Identification and Antibiotic Susceptibility Testing

The bacterial species-level identification and antibiotic susceptibility testing of the isolates were performed with automated microbiological systems (Vitek2 automated system) at the Bahrain Defense Force Hospital and Bahrain Specialist Hospital and a BD Phoenix^TM^ automated system at the Al-Salmaniya Medical Complex and King Hamad University Hospital. Only the isolates that were identified as *A. baumannii* resistant to carbapenems were included for further molecular analysis. As per each hospital’s antibiotic policies, the isolates were tested against certain antibiotics. The tested and non-tested antibiotics are presented in Figure 1A.

### 4.3. Amplification of Carbapenemases Genes via Polymerase Chain Reaction

For carbapenemase gene detection, the DNA of the bacterial strains was extracted from the CRAB pure culture using the boiling method [29]. The PCR reactions were carried out in a total volume of 25 μL, consisting of 12.5 μL of PCR Master Mix, 9 μL of DNase/RNase-free water, 0.5 μL each of forward and reverse primer, and 2.5 μL of DNA template. Each primer specific for the carbapenemase genes (Class A: *bla_KPC_*; Class B: *bla_IMP_***,**
*bla_NDM_*, and *bla_VIM_*; Class D: *bla_OXA-23_*, *bla_OXA-24_*, *bla_OXA-40_*, *bla_OXA-48_*, *bla_OXA-51_*, and *bla_OXA-58_*) had a specific thermal cycle that was optimized separately. The amplicons were detected via 1.5% gel electrophoresis, and the bands were visualized under UV illumination [22]. The positive quality control strains used were multidrug-resistant *A. baumannii* ATCC 19606 and *Klebsiella pneumoniae*. The specific primers and the cycling conditions used in the study are shown in Table 3.

## 5. Conclusions

This study provides a clear picture of the currently prevalent bla-carbapenemases in the Kingdom of Bahrain. Oxacillinases (Class D) were the predominant carbapenemases; the most common genes detected were *bla_OXA-51_* and *bla_OXA-23_*. From Class B, *bla_IMP_* was also detected at a significantly higher percentage. The presence of other Class B genes (such as *bla_NDM_* and *bla_VIM_*) and Class A genes (*bla_KPC_*), though in smaller percentages, is quite alarming. The rate of resistance to most antibiotics is high in our region. These results emphasize the significance of rational antibiotic therapy and ongoing stringent surveillance and infection control strategies to successfully curb the spread of these clinical strains.

## Figures and Tables

**Figure 1 antibiotics-12-01198-f001:**
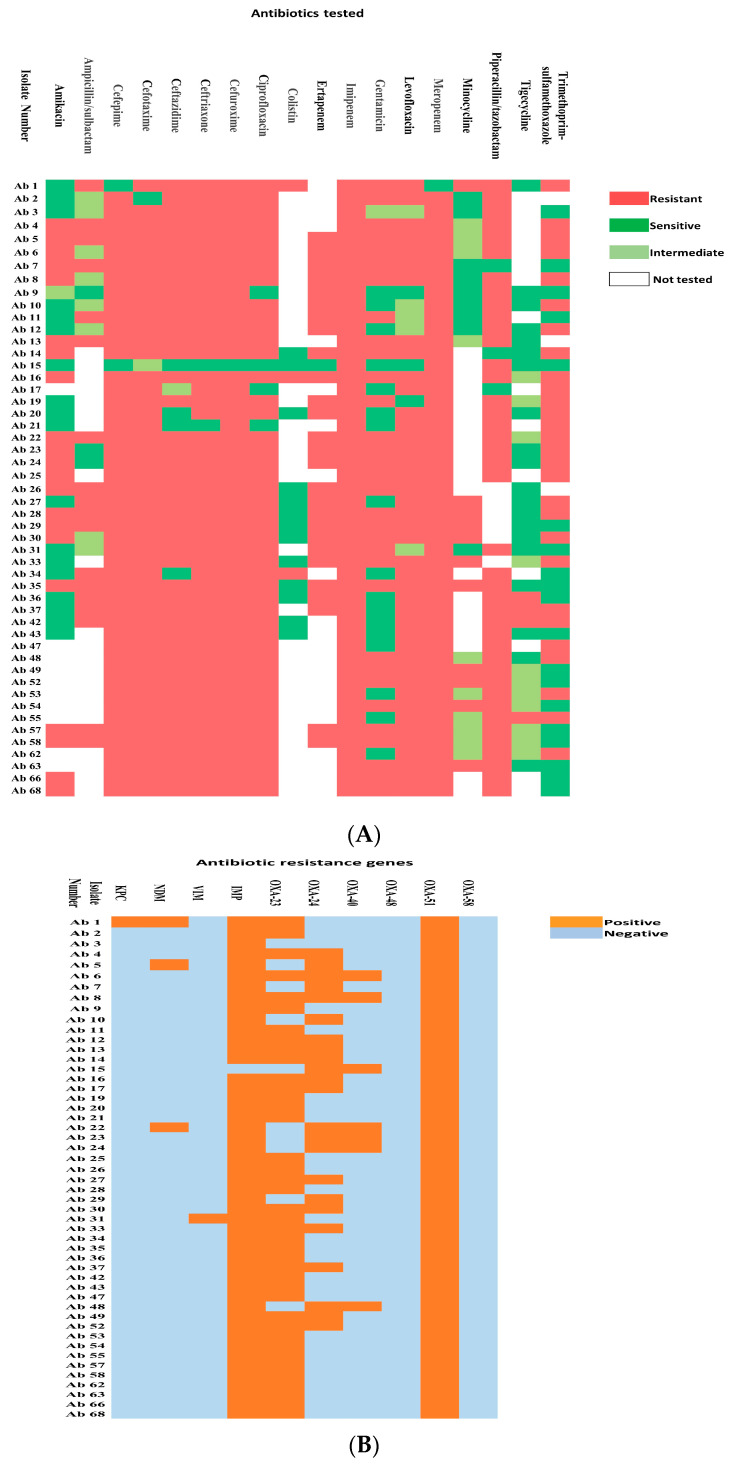
(**A**) The antibiotic resistance profile of CRAB isolates. (**B**) The presence of carbapenemases genes in CRAB isolates.

**Figure 2 antibiotics-12-01198-f002:**
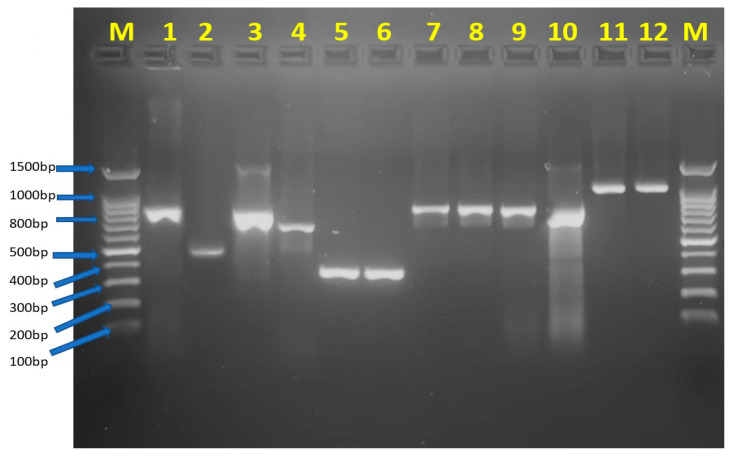
A representative PCR gel (1.5% agarose) showing respective *bla*-carbapenemase genes. Lane M denotes the molecular weight marker, lane 1 shows representative *bla*_KPC_ (881 bp), lane 2—*bla*_IMP_ (484 bp), lane 3—*bla*_NDM_ (825 bp), lane 4—*bla*_VIM_ (601 bp), lanes 5 and 6—*bla*_OXA-51_ (353 bp), lanes 7, 8, and 9—*bla*_OXA-23_ (821 bp), lane 10—*bla*_OXA-24_ (809 bp), and lanes 11 and 12—*bla*_OXA-40_ (1024 bp).

**Table 1 antibiotics-12-01198-t001:** The frequency of resistance of the respective antibiotics in the CRAB isolates.

Antibiotics	Total No. of Isolates Tested	Resistant Isolates (%)	Intermediate Isolates (%)
Amikacin	40	21 (52.5%)	1 (2.5%)
Ampicillin/sulbactam	33	22 (66.6%)	8 (24.2%)
Cefepime	50	48 (96%)	
Cefotaxin	50	48 (96%)	1 (2%)
Ceftazidime	50	45 (90%)	1 (2%)
Ceftriaxone	50	48 (96%)	
Cefuroxime	50	49 (98%)	
Ciprofloxacin	50	46 (92%)	
Colistin	18	5 (27.7%)	
Ertapenem	28	27 (96.4%)	
Imipenem	50	50 (100%)	
Gentamicin	50	32 (64%)	1 (2%)
Levofloxacin	48	42 (87.5%)	3 (6.25%)
Meropenem	50	49 (98%)	
Minocycline	30	11 (36.6%)	10 (33.3%)
Piperacillin/tazobactam	44	43 (97.7%)	
Tigecycline	35	3 (5.5%)	12 (34.2%)
Trimetoprim-sulfamethoxazole	47	28 (59.5%)	

**Table 2 antibiotics-12-01198-t002:** Frequency and pattern of the combinations of carbapenem resistance genes.

Combination of Carbapenem-Resistance Genes	Number of Isolates (*n* = 50)
*bla_OXA-51,_ bla_OXA-23,_ bla_OXA-24,_ bla_OXA-40,_ bla_IMP_*	2
*bla_OXA-51,_ bla_OXA-23,_ bla_OXA-24,_ bla_IMP_*	12
*bla_OXA-51,_ bla_OXA-23,_ bla_IMP_*	22
*bla_OXA-51,_ bla_OXA-23,_ bla_OXA-40,_ bla_IMP_*	3
*bla_OXA-51,_ bla_OXA-23,_*	2
*bla_OXA-51,_ bla_OXA-24,_ bla_IMP_*	3
*bla_OXA-51,_ bla_OXA-23,_ bla_IMP,_ bla_VIM_*	1
*bla_OXA-51,_ bla_OXA-23,_ bla_OXA-24,_ bla_KPC,_ bla_IMP,_ bla_NDM_*	1
*bla_OXA-51,_ bla_OXA-24,_ bla_OXA-40,_ bla_IMP,_ bla_NDM_*	1
*bla_OXA-51,_ bla_OXA-24,_ bla_IMP,_ bla_NDM_*	1
*bla_OXA-51,_ bla_OXA-24,_ bla_OXA-40_*	1
*bla_OXA-51,_ bla_IMP_*	1

**Table 3 antibiotics-12-01198-t003:** List of carbapenemase-gene-specific primers used in this study.

Genes Targeted	Primer Sequence (5′→3′)	Amplicon Size (bp)	Reference
KPC	F-ATGTCACTGTATCGCCGTCT R-TTACTGCCCGTTGACGCCCA	881	[20]
VIM	F-ATTCCGGTCGG(A=G) GAGGTCCG R-TGTGCTKGAGCAAKTCYAGACCG	601	[20]
NDM	F-GGCCGTATGAGTGATTGC R-GAAGCTGAGCACCGCATTAG	825	[20]
IMP	F-CGGCC(G=T) CAGGAG(A=C) G(G-T) CTTT R-AACCAGTTTTGC(C=T) TTAC(C=T) AT	484	[20]
OXA-23	F-ATGAATAAATATTTTACTTG RTTAAATAATATTCAGCTGTT	821	[20]
OXA-24	F-ATACTTCCTATATTCAGCAT R-GATTCCAAGATTTCTAGCG	809	[20]
OXA-40	F-GTACTAATCAAAGTTGTGAA R-TTCCCCTAACATGAATTTGT	1024	[30]
OXA-48	F-GCTTGATCGCCCTCGATT R-GATTTGCTCCGTGGCCGAAA	281	[20]
OXA-51	F-TAATGCTTTGATCGGCCTTG R-TGGATTGCACTTCATCTTGG	353	[20]
OXA-58	F-ATGAAATTATTAAAAATATTGAGT R-ATAAATAATGAAAAACACCCAA	840	[20]

## Data Availability

We confirm that all the data are presented in this article.

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
