# Peer review of "Prevalence of Carbapenemases in Carbapenem-Resistant Acinetobacter baumannii Isolates from the Kingdom of Bahrain"

_antibiotics, 2023, doi:10.3390/antibiotics12071198_

Round 1

Reviewer 1 Report

Nouf Al-Rashed et al. have conducted a study investigating the prevalence of carbapenemases in carbapenem-resistant Acinetobacter baumannii (CRAB) isolates from the Kingdom of Bahrain. The authors found that class D carbapenemases were more frequently detected in CRAB isolates, utilizing conventional PCR and antibiotic susceptibility methods. Based on my evaluation, I have a few constructive comments for further enhancing the study, and I strongly recommend its publication.

Comments:

·      Line 74 write the composition of the glycerol milk. For example, percentage of glycerol and milk or the composition of milk

·      Line 112 remove the before isolates word line 113

·      Figure 2 it will be easy to understand if the expected size of the is written in the gel or the figure legend.

Comments:

·      Line 74 write the composition of the glycerol milk. For example, percentage of glycerol and milk or the composition of milk

·      Line 112 remove the before isolates word line 113

·      Figure 2 it will be easy to understand if the expected size of the is written in the gel or the figure legend.

Author Response

We have made the following point-by-point responses to the reviewers’ comments. We have indicated where/how the manuscript has been revised to address the comments (highlighted in yellow color).

Reviewer 2 Report

In this manuscript, the authors examined the presence of bla-carbapenemase genes in a collection of carbapenem-resistant Acinetobacter baumannii (CRAB) isolates, and tested their antibiotic susceptibility. The authors find that blaIMP and blaOXA-51 exist in almost all the isolates, in various combinations with several other carbapenemase-producing genes. Their results also show that the isolates are extensively drug resistant and are most susceptible to tigecycline. Overall, this is a well conducted and written study. I don’t have any experimental requests, but I have a few questions below out of my own curiosity:

1.     In Figure 1, there’re a bunch of “not tested” blocks. Is there a technical reason why those are not tested? If so, I would suggest adding a note about this in the Methods section.

2.     The legends/labels in Figure 1 are difficult to read. Please use a higher-resolution image or increase the font size.

3.     Have the authors tried to use clustering methods to cluster the rows in Figure 1? I’d be interested to see if that would reveal any correlation between the combinations of carbapenemase genes and the antibiotic-resistance pattern.

Author Response

(The authors gave the same response as above.)

Reviewer 3 Report

I reviewed  "Prevalence of carbapenemases in carbapenem-resistant Acinetobacter baumannii isolates from the Kingdom of Bahrain", an interesting article of more epidemiological interest.

AB resistance to carbapenems is not a recent or not at all known topic, but determination of resistance genes is newly introduced.

My observations are:

1. Restructure the abstract

2.Figure 1 contains two aspects, resistance profile and genes; can be separate? It is very difficult to read it

3.Add tables to contain the data from the text, resistance % different tested antibiotics

3. Add data about the importance of the subject

Author Response

(The authors gave the same response as above.)

Round 2

Reviewer 3 Report

Dear authors

You provided a new and improved paper.